# Targeting Microglia/Macrophages Notch1 Protects Neurons from Pyroptosis in Ischemic Stroke

**DOI:** 10.3390/brainsci13121657

**Published:** 2023-11-29

**Authors:** Ran Chen, Hua Zhu, Zhihui Wang, Yonggang Zhang, Jin Wang, Yingao Huang, Lijuan Gu, Changyong Li, Xiaoxing Xiong, Zhihong Jian

**Affiliations:** 1Department of Neurosurgery, Renmin Hospital of Wuhan University, Wuhan 430064, China; whuchenran@163.com (R.C.); 2019203020043@whu.edu.cn (H.Z.); 2016302180028@whu.edu.cn (Z.W.); 2015302180367@whu.edu.cn (Y.Z.); huangyingao@whu.edu.cn (Y.H.); xiaoxingxiong@whu.edu.cn (X.X.); 2Department of Anesthesia, Renmin Hospital of Wuhan University, Wuhan 430064, China; wang.jin@whu.edu.cn; 3Central Laboratory, Renmin Hospital of Wuhan University, Wuhan 430064, China; gulijuan@whu.edu.cn; 4Department of Physiology, School of Basic Medical Sciences, Wuhan University, Wuhan 430072, China; lichangyong@whu.edu.cn

**Keywords:** ischemic stroke, pyroptosis, neuroinflammation, Notch1, CIRI

## Abstract

Background and Aims: The immune-inflammatory cascade and pyroptosis play an important role in the pathogenesis of cerebral ischemia-reperfusion injury (CIRI). The maintenance of immune homeostasis is inextricably linked to the Notch signaling pathway, but whether myeloid Notch1 affects microglia polarization as well as neuronal pyroptosis in CIRI is not fully understood. This study was designed to clarify the role of myeloid Notch1 in CIRI, providing new therapeutic strategies for ischemic stroke. Methods and Results: Myeloid-specific Notch1 knockout (Notch1^M-KO^) mice and the floxed Notch1 (Notch1^FL/FL^) mice were subjected to middle cerebral artery occlusion (MCAO). After 3 days of CIRI, we evaluated the neurological deficit score and cerebral infarction volume. Immunofluorescence staining was used to detect the expression of Notch1 and microglial subtype markers. Cerebral infiltrating macrophages were detected by flow cytometry. RT-qPCR was used to detect pro-inflammatory cytokines. Western blot was used to detect the expression of pyroptosis related proteins. The Notch1-siRNA transfected BV2 cells were co-cultured with HT22 cells to investigate the potential mechanisms by which microglial Notch1 affects neuronal pyroptosis induced by anoxia/reoxygenation in vitro. We found that Notch1 was activated in cerebral microglia/macrophages after CIRI. Myeloid Notch1 deficiency decreased the cerebral infarct volume (24.17 ± 3.29 vs. 36.17 ± 2.27, *p* < 0.001), neurological function scores (2.33 ± 0.47 vs. 3.17 ± 0.37, *p* < 0.001) and the infiltration of peripheral monocytes/macrophages (3.26 ± 0.53 vs. 5.67 ± 0.57, *p* < 0.01). Strikingly, myeloid-specific Notch1 knockout alleviated pyroptosis. Compared with microglia M1, increased microglia M2 were detected in the ischemic penumbra. In parallel in vitro co-culture experiments, we found that Notch1 knockdown in microglial BV2 cells inhibited anoxia/reoxygenation-induced JAK2/STAT3 activation and pyroptosis in hippocampal neuron HT22 cells. Conclusions: Our findings elucidate the underlying mechanism of the myeloid Notch1 signaling pathway in regulating neuronal pyroptosis in CIRI, suggesting that targeting myeloid-specific Notch1 is an effective strategy for the treatment of ischemic stroke.

## 1. Introduction

Stroke is the main cause of death and physical disability worldwide, and over 80% of stroke cases are ischemic due to cerebral artery occlusion [1,2]. Among the current clinical methods for treating ischemic stroke, revascularization and unblocking the occluded blood vessels is commonly used to treat acute ischemic stroke. However, this treatment can also lead to ischemia/reperfusion injury [3]. Ischemic tissue can be divided into irreversibly damaged tissue in the infarct core and surrounding tissue in the ischemic penumbra. The peri-infarct dead zone contains tissue to which blood flow is reduced; this tissue can be salvaged through early reperfusion [4]. If reperfusion of ischemic brain tissue is delayed, the infarct core may extend into the penumbra area. Previous studies have shown that microglia was activated by the inflammatory response following ischemia, as well as macrophage infiltration, and these are key factors in ischemic stroke [5]. In the earliest stage of cerebral ischemia, microglia and infiltrating macrophages exhibit a neuroprotective anti-inflammatory phenotype before being transformed to a proinflammatory phenotype [6,7]. Inhibiting the polarization of microglia toward the proinflammatory phenotype or promoting their polarization toward the anti-inflammatory phenotype has certain neuroprotective effects [8]. Therefore, transforming microglia/macrophages from the proinflammatory to the anti-inflammatory phenotype is a key strategy for the treatment of ischemic stroke and is also crucial for preserving the function of neurons in the ischemic penumbra.

Currently, it is believed that neuronal death occurs through multiple mechanisms of cell death after ischemic stroke. In recent years, an increasing number of cell death mechanisms, including a new type of programmed cell death called pyroptosis, have been discovered [9]. Pyroptosis is caspase-1 dependent and is related to the release of proinflammatory cytokines [10]. Caspase-1 is activated by various inflammasome complexes in innate immunity, including NLRP3 [11]. Then, Cl-Caspase1 cleaves Gasdermin D (GSDMD) to form pores that directly permeabilize the plasma membrane, leading to rupture of the plasma membrane and release of inflammatory cellular contents, including mature IL-1β and IL-18 [12,13], which induces an inflammatory response or initiates the inflammatory cell death process [14,15,16]. These cytokines are released into the extracellular environment, thus causing toxicity to neurons. 

Notch is a membrane receptor that regulates basic cellular physiological functions such as cell proliferation, differentiation and development. The Notch intracellular structural domain (NICD) is the active intracellular portion of Notch. The function of this active intracellular fraction is to translocate to the nucleus to interact with transcription factors and ultimately regulate downstream gene expression [17]. Existing studies suggest that Notch1 antisense transgenic mice exert beneficial effects on cerebral ischemia-induced neurological dysfunction and cerebral infarction. γ-secretase-mediated Notch1 activation impairs neuronal function after ischemic stroke by increasing apoptosis, proinflammatory leukocyte infiltration, and microglial activation [18]. However, the exact mechanism by which microglia Notch1 is involved in neuronal pyroptosis induced by anoxia/reoxygenation is not known.

This study aims to investigate the specific role of myeloid Notch1 in CIRI. By constructing an MCAO model in myeloid Notch1 knockout mice, we found that myeloid Notch1 knockout has a protective effect on CIRI. By using siRNA virus transfection technology and in vitro cells co-culture, it is confirmed that BV2 cells with a low expression of Notch1 have an inhibitory effect on OGD/R induced neuronal pyroptosis. Our findings provide promising intervention targets for the development of new treatment strategies for ischemic stroke in the future.

## 2. Materials and Methods

### 2.1. Animals

Floxed Notch1 (Notch1^FL/FL^) mice were used to generate myeloid-specific Notch1 knockout (Notch1^M-KO^) mice along with myeloid-specific mice expressing cre recombinase (LysM-Cre), both from Jackson Laboratory. Eight weeks male Notch1^FL/FL^ and Notch1^M-KO^ mice were used in this study. They were kept in a standard environment at the Animal Experiment Center of Renmin Hospital of Wuhan University. The mice were kept at a constant humidity and temperature (20–24 °C). The standard rearing environment was created by a cycle of half daylight and half nighttime. All animal experiments were performed based on the Animal Ethics Committee, Renmin Hospital of Wuhan University, China (1 August 2022, permit No: WDRM20220703A).

### 2.2. Middle Cerebral Artery Occlusion (MCAO) and Reperfusion

According to the formerly described model, the cerebral ischemia/reperfusion model was established using 8 weeks male C57BL/6J mice, Notch1^FL/FL^ mice and Notch1M-KO mice in in vivo experiments [19]. Briefly, under isoflurane anesthesia, a midline vertical incision was made along the mouse’s neck. Using standardized 6.0 nylon monofilament insertion into the left common carotid artery, the origin of MCA was ultimately blocked by advancing through the internal carotid artery. After 1 h of occlusion, the nylon monofilament thread was removed to induce reperfusion. The same surgery, without the insertion of the thread, was performed on the sham group mice.

### 2.3. Neurological Function Assessment

Neurologic grading of focal cerebral ischemia/reperfusion injury (CIRI) was determined by a 5-point neurologic deficit scale (0, no symptoms of nerve damage; 1, failure to extend the contralateral paw; 2, leaning to the contralateral side without circling; 3, falling to the contralateral side with unidirectional circling; 4, inability to walk spontaneously); neurological function was assessed in a blinded manner [19].

### 2.4. Infarct Area Measurement

Mice were euthanized (with an overdose of isoflurane) 72 h after cerebral ischemia-reperfusion. The dissected brains were immersed in PBS (4 °C) for 15 min before being cut into 2 mm coronal sections with a tissue slicer. The sections were incubated in 2% 2,3,5-triphenyltetrazolium chloride (TTC) for 15 min at the constant temperature of 37 °C. Digital images of the stained sections were taken and analyzed. The infarct area was distinguished and labeled. The area of infarction in each section was quantified using ImageJ software version 6.1. The area of infarction was determined by subtracting the area of undamaged tissue in the left hemisphere from that in the normal contralateral hemisphere to control for cerebral edema. The area of infarction in each brain section was determined in a blinded manner, and the calculated infarct area is expressed as a percentage of the area of the ipsilateral hemisphere.

### 2.5. Immunofluorescence Staining

Samples were collected in the same manner as for cerebral infarct area measurement. Afterward, the brain slices fixed in 4% Paraformaldehyde for 10 min were treated with 0.3% Triton X-100 for 10 min. Finally, in order to block non-specific binding, 10% bovine serum albumin (BSA) was used to incubate the slices at room temperature for 1 h. The slices were washed three times using PBS (10 min each wash). Finally, they were incubated overnight at 4 °C with primary antibodies against CD68 (diluted 1:200, 14-0688-82, Invitrogen, Waltham, MA, USA), Notch1 (diluted 1:200, MA5-32080, Invitrogen), Ym1 (diluted 1:200, 60130, STEMCELL Technologies, Vancouver, BC, Canada), iNOS (diluted 1:200, GB11119-100, Servicebio, Wuhan, China) and Iba1 (diluted 1:500, GB12105, Servicebio, Wuhan, China). After washing the incubated primary antibody, the sections were incubated in the secondary antibody for 1 h at room temperature. The secondary antibodies are as follows, Alexa Flour 594 Donkey anti rabbit IgG antibody (diluted 1:200, ANT030, Ant Gene, Wuhan, China), Alexa Flour 594 Donkey anti mouse IgG antibody (diluted 1:200, ANT029, Ant Gene, Wuhan, China), Alexa Fluot 488 Goat anti mouse IgG antibody (diluted 1:200, ANT044, Ant Gene, Wuhan, China), Alexa Fluot 488 Donkey anti rabbit IgG antibody (diluted 1:200, ANT024, Ant Gene, Wuhan, China). After incubation, the samples were placed under a fluorescence microscope (Olympus Optical, Tokyo, Japan) for observation. Five different regions of the ischemic penumbra of each mouse brain slice were assessed (five mice per group). Evaluators used ImageJ to blindly determine the number of immunoreactive positive cells in the regions of interest.

### 2.6. Western Blot Analysis

Total protein was harvested from HT-22 cells and ipsilateral brain tissues of mice subjected to different treatments. The homogenized tissues were mixed into protease and phosphatase inhibitors (Servicebio), and then pre-cooled RIPA buffer (Beijing Applygen Technologies Inc., Beijing, China) was added and lysed thoroughly. An appropriate amount of the prepared protein sample was added to the gel, separated by electrophoresis, and then transferred to a PVDF membrane (0000151028, EMD Millipore, Beijing, China). The membranes were then incubated for 1 h at 23 °C in blocking buffer (5% skim milk in 20 mmol/L Tris-HCl (pH 7.5), 137 mmol/L NaCl, and 0.2% Tween-20). The membranes were then incubated overnight at 4 °C with primary antibodies specific for Notch1 (diluted 1:1000; MA5-32080, Invitrogen), cleaved caspase1 (diluted 1:1000; 89332, Cell Signaling Technology, Danvers, MA, USA), cleaved GSDMD (diluted 1:1000; P30823S, ABMART, Shanghai, China), NLRP3 (diluted 1:1000; T55651S, ABMART, Shanghai, China), IL1β (diluted 1:1000; 12242, Cell Signaling Technology), IL18 (diluted 1:1000; M027287S, ABMART), JAK2 (diluted 1:1000; 3230, Cell Signaling Technology, Danvers, MA, USA), P-JAK2 (diluted 1:1000; 66245, Cell Signaling Technology), STAT3 (diluted 1:1000; 12640S, Cell Signaling Technology Danvers, MA, USA), P-STAT3 (diluted 1:1000; 9145S, Cell Signaling Technology, Danvers, MA, USA), Bax (diluted 1:1000; 41162, Cell Signaling Technology, Danvers, MA, USA), Bcl2 (diluted 1:1000; sc-492, Santa Cruz Biotechnology, Dallas, TX, USA), iNOS (diluted 1:1000; ab49999, Abcam, Cambridge, MA, USA), GAPDH (diluted 1:1000; GB11002, Servicebio, Wuhan, China) and β-actin (diluted 1:1000; GB12001, Servicebio, Wuhan, China). The membranes were washed three times with pre-prepared TBST, and then incubated with HRP-conjugated secondary antibody for 1 h. The secondary antibodies are as follows, Anti rabbit IgG HRP-linked antibody (diluted 1:5000, 7074S, Cell Signaling), Anti mouse IgG HRP-linked antibody (diluted 1:5000, 7076S, Cell Signaling Technology, Danvers, MA, USA). The Bio-Rad Imaging System was used to capture the protein bands. The optical density of each target protein band was normalized to the optical density of β-actin or GAPDH bands and analyzed by ImageJ software to finalize the relative expression of each protein.

### 2.7. Flow Cytometry Analysis

After 72 h of cerebral ischemia, mice euthanized with an overdose of isoflurane were subjected to cardiac perfusion by utilizing pre-cooled saline in advance. The brain tissue was isolated, the cerebellum and brainstem were removed, and the ischemic hemisphere was sufficiently minced and completely immersed in 2 mL of digestion solution (DMEM containing 1 mg/mL collagenase IV and 1 mg/mL DNAase I). The above mixture was digested by shaking (180 r/min) in a constant temperature shaker at 37 °C. Digestion was terminated by placing the sample on ice after 45 min. The digested brain tissue suspension was passed through a 40 µm cell filter to remove insufficiently digested tissue masses and cell clusters. The filtrate was centrifuged for 5 min (1200 r/min speed), the supernatant was removed and the precipitate was composed of brain cells. The precipitate was resuspended in 4 mL of 30% Percoll solution to obtain a single-cell suspension. Then, 3.5 mL of 70% Percoll solution was slowly added to a 15 mL centrifuge tube, 3.5 mL of 37% Percoll solution was slowly poured on top, and 4 mL of monocyte suspension was added as the uppermost layer. After centrifugation, a white flocculent layer (single nucleated cells, including lymphocytes) was observed at the delamination of the 70% and 37% Percoll solutions; the white flocculent layer was collected with a pipette and centrifuged at 1200 r/min for 10 min. Cells were washed with FACS buffer and incubated with CD45 (553079, BD Bioscience, Franklin Lakes, NJ, USA) and CD11b antibodies (550993, BD Bioscience) (protected from light and frozen for 30 min). CytoFLEX flow cytometer (Beckman Coulter, Brea, CA, USA) was used for analysis. The fluorescence channels corresponding to CD45 (FITC) and CD11b (PerCP-Cy5.5) were selected. A new FSC-SSC scatter plot was created, giving each sample a settled name (including the blank tube). The *X*-axis and *Y*-axis maximum settings were adjusted so that the cells were in the appropriate position in the center of the scatter plot. Each gate of the target cells group was circled and applied to the histogram. The compensation was adjusted appropriately. About 100,000 cells were collected in each group for analysis on CytoFLEX flow cytometer (Beckman Coulter). The expression of target molecules was analyzed using (v2.3, Beckman Coulter) and the final results were processed using FlowJo_V10 software.

### 2.8. Quantitative Real-Time PCR

TRIzol reagent (Invitrogen) was used to extract total RNA from ischemic penumbra tissue and HT22 cells according to the instructions of the reagents. Hifair^®^ Ⅱ 1st Strand cDNA Synthesis SuperMix (gDNA Digester Plus) (11123ES, Yeasen Biotechnology, Shanghai, Co., Ltd., Shanghai, China) was used to perform reverse transcription. The primers listed in Table 1 were used to amplify the cDNA. SYBR Premix Ex Taq2 (TaKaRa, Osaka, Japan) used for real-time fluorescent quantitative PCR was mixed with synthetic primers and the cDNA. Target mRNA levels were normalized to GAPDH levels.

### 2.9. PPI (Protein–Protein Interaction) Network Analysis

The functional protein interaction database—STRING database (https://cn.string-db.org/)—is a database that includes all types of known and predicted protein interactions [20]. Protein names were entered into the box, and the organism set to *Mus musculus*. PPI networks with moderate confidence scores ≥ 0.4 were statistically generated. Then, Cytoscape software (version 3.7.2) was employed to merge the PPI networks of Notch1, JAK2 and STAT3.

### 2.10. In Vitro Transfection

A lentivirus was produced by Shanghai Genechem Co., Ltd., (Shanghai, China) The target sequence of the Notch1-targeted RNAi was GCCAGGTTATGAAGGTGTATA. BV2 cells were suspended in complete medium at a density of 3~5 × 10^4^ cells/mL, 2 mL of the cell suspension was plated in each well of a six-well plate (the volume was 1 mL at the time of transfection), and the cells were incubated at 37 °C and 5% CO_2_ for 24 h until they reached 20–30% confluence. Following the protocol, 10 μL of virus solution and 40 μL of HiTransG A infection solution were added to each well. The cells were incubated in the 37 °C and 5% CO_2_ incubator for 16 h. Then, the transfection solution was removed and replaced with a complete culture medium. The cells were then incubated for 72 h. After 72 h of transfection, the transfection efficiency was observed under a fluorescent inverted microscope, and the appropriate working concentration (2 μg/mL) of puromycin was added for screening. The cells were screened until all the cells in the control groups died, and then the puromycin concentration was reduced to 1 μg/mL. The transfected cells subsequently screened and amplified. Then, the successfully transfected cells were collected for qPCR and Western blot. Notch1-siRNA transfected BV2 cells as well as NC-siRNA transfected BV2 cells (negative control) were frozen for subsequent experiments.

### 2.11. Co-Culture Experiment

HT22 cells were plated at a density of 1 × 10^4^ cells/cm^2^ in the lower chamber of a 6-well Transwell plate (polycarbonate membrane with a pore size of 0.4 μm), and stably transfected BV2 cells were plated in the top chamber of the Transwell plate at the same density (1 × 10^4^ cells/cm^2^). A control 6-well Transwell plate in which HT22 cells and BV2 cells were plated alone at a density of 1 × 10^4^ cells/cm^2^ was also prepared. The cells were cultured for 24 h in a 37 °C and 5% CO_2_ incubator. In the control group, we used high-glucose medium (C11995500BT, Dulbecco’s Modified Eagle Medium, Gibco, Shanghai, China), with 10% FBS (11011-8611, TianHang, Biotechnology Co., Ltd., Huzhou, China). In OGD/R group, we used glucose-free medium (PM150270, Dulbecco’s Modified Eagle Medium, Procell Life Science & Technology Co., Ltd., Wuhan, China).

### 2.12. In Vitro OGD/R Model

Cells were subjected to OGD/R when they were in the logarithmic growing period. The initial medium was removed and the cells in the upper and lower chambers were washed twice with PBS. The medium of the experimental cells was replaced with glucose-free DMEM, whereas the blank control cells were left untreated. Subsequently, the cells co-cultured in the Transwell plates were transferred to an anoxic incubator containing 1% O_2_, 5% CO_2_ and 94% N_2_ and incubated at 37 °C for 6 h to simulate OGD. The glucose-free medium (PM150270, Dulbecco’s Modified Eagle Medium, Procell Life Science & Technology Co., Ltd., China) was replaced by high-glucose medium (C11995500BT, Dulbecco’s Modified Eagle Medium, Gibco, China) with 10% FBS (11011-8611, TianHang, Biotechnology Co., Ltd., China), and the cells were cultured in a normal culture environment for another 12 h to simulate reperfusion.

### 2.13. Cell Counting Kit-8 Assay

HT22 cells were transfected with NC-siRNA or Notch1-siRNA for 24 h and cultured under normal conditions or subjected to OGD/R. The HT22 cells were isolated after 24 h of incubation and plated in 96-well plates containing complete medium. CCK-8 reagent (10 μL) was added to each well of medium and the cells were incubated at 37 °C for 1.5 h protected from light. Absorbance values were measured at 450 nm using an enzyme labeler.

### 2.14. Annexin V-EGFP Apoptosis Detection

The cell culture was first aspirated into a centrifuge tube, followed by gentle washing of the lower chamber HT22 cells once with pre-cooled PBS (pH 7.4) and addition of an appropriate amount of trypticase cell digest (without EDTA). Incubate at room temperature until gentle blowing can dislodge the adherent cells, and aspirate the trypticase cell digest. Add the above culture solution to the cells and mix slightly, transfer to a centrifuge tube, centrifuge (2000× *g*, 1 min), discard the supernatant, collect the cells, gently resuspend the cells with pre-cooled PBS (pH 7.4) and count the cells. Add 4 µL of Annexin V-EGFP and 4 µL of propidium iodide (PI) to 192 µL of binding buffer, mix thoroughly and centrifuge briefly to form a Staining Buffer. Take 0.5 × 10^5^~1 × 10^5^ cells, centrifuge (2000× *g*, 1 min), discard supernatant, add 200 µL of Staining Buffer and gently resuspend cells at room temperature. The cells were gently resuspended and incubated for 5~10 min at room temperature or 37 °C, protected from light, and immediately analyzed by flow cytometry.

### 2.15. Statistical Analysis

GraphPad Prism software (version 7.03) was used for statistical analysis. Data results were expressed as the mean ± SD. The data were analyzed by Student’s *t* test for two group comparisons or one-way analysis of variance (ANOVA), followed by Dunnett’s post hoc test for multiple comparisons. *p* values less than 0.05 were considered to indicate statistical significance.

## 3. Results

### 3.1. Microglia Notch1 Expression Is Elevated in the Ischemic Penumbra in CIRI

To investigate whether Notch1 plays the role as a regulatory hub during the acute phase of cerebral ischemia–reperfusion injury and whether it affects neuronal pyroptosis after stroke, we examined the protein expression of Notch1 and NLRP3 in the ischemic penumbra of mice on day 3 after MCAO. The expression of Notch1 (0.81 ± 0.05 vs. 0.46 ± 0.07, *p* < 0.001) and NLRP3 (0.70 ± 0.05 vs. 0.42 ± 0.04, *p* < 0.001) was elevated in the ischemic penumbra in the MCAO group (Figure 1A,B). The expression of Notch1 and NLRP3 genes increased when detected by qPCR (Figure 1C). Using double immuno-fluorescence staining, we found that the increased expression of Notch1 was primarily localized in microglia/macrophages of ischemic penumbra area (Figure 1D,E). Consistent with this result, BV2 cells were subjected to OGD/R modeling by in vitro experiments, and the expression of Notch1 and NLRP3 proteins was detected by using Western blot. The results showed that OGD/R treatment increased the expression of Notch1 (0.61 ± 0.05 vs. 0.35 ± 0.06, *p* < 0.001) and NLRP3 (0.68 ± 0.06 vs. 0.39 ± 0.04, *p* < 0.001) in BV2 cells compared with the control group (Figure 1F,G). To further validate the above results, we subjected HT22 neurons to reoxygenation after OGD (6 h) to construct a model of OGD/R in vitro and then measured the expression levels of Notch1 and NLRP3 in each group. Compared with the control group, the expression levels of protein Notch1 (0.69 ± 0.05 vs. 0.43 ± 0.03, *p* < 0.001) and NLRP3 (0.70 ± 0.04 vs. 0.44 ± 0.04, *p* < 0.001) sensibly rose in the OGD/R group (Figure 1H,I); qPCR was conducted to verify the variation in the gene expression level, and the results were consistent with those reported above (Figure 1J).

### 3.2. Notch1 Knockout Attenuates CIRI and Cerebral Migration of Monocyte-Derived Macrophages

To investigate whether Notch1 in myeloid cells occupies a pivotal position in cerebral ischemia-reperfusion injury, we generated Notch1^M-KO^ (myeloid-specific Notch1 knockout) and Notch1^FL/FL^ mice. To investigate whether knocking out Notch1 in the myeloid system affects the development of myeloid cells, we conducted flow cytometry analysis on bone marrow cells from male 8-week-old WT, Notch1^FL/FL^ mice and Notch1^M-KO^ mice. The results showed that there was no statistically significant difference in the proportion of T cells, B cells, neutrophils and macrophages in the total live cells extracted from the bone marrow of the three groups of mice (Appendix A). Furthermore, we verified the absence of Notch1 in myeloid cells of Notch1^M-KO^ mice by Western blot (Appendix A). In this study, an MCAO model was constructed to assess the direct effect of Notch1 on CIRI in Notch1^M-KO^ mice and Notch1^FL/FL^ mice, and sham-pseudosurgical mice were constructed as blank controls. The neurological function score of Notch1^M-KO^ mice measured on the third day after reperfusion was lower than Notch1^FL/FL^ mice (2.33 ± 0.47 vs. 3.17 ± 0.37, *p* < 0.001). The infarct volume of Notch1^M-KO^ mice was smaller than that of Notch1^FL/FL^ mice (24.17 ± 3.29 vs. 36.17 ± 2.27, *p* < 0.001). (Figure 2A–C). Using immunofluorescence staining, we found that myeloid-specific Notch1 knockout significantly reduces total microglia/macrophages in the central ischemic penumbra (Figure 2E). The number of Iba1^+^ cells notably decreased in the ischemic penumbra in Notch1^M-KO^ mice compared with Notch1^FL/FL^ mice after MCAO (56.00 ± 4.69 vs. 108.00 ± 8.11, *p* < 0.001) (Figure 2D). Macrophages in the brain include monocyte-derived macrophages (MDMs) and microglia-derived macrophages (MiDMs)/brain-resident microglia. We performed flow cytometry and found that the number of MDMs (CD45^high^CD11b^+^) diminished in the Notch1^M-KO^ mice compared with the Notch1^FL/FL^ mice (3.26 ± 0.53 vs. 5.67 ± 0.57, *p* < 0.01) (Figure 2F,G), which indicated that myeloid-specific Notch1 knockout effectively inhibited the invasion of peripheral macrophages into the cerebral ischemic areas.

### 3.3. Notch1 Knockout Promotes Microglial Polarization toward M2 Phenotypes in the Ischemic Penumbra

Microglia and macrophages are rapidly activated after ischemia, and they are important members of the immune cascade after acute ischemic events. Activated macrophages are converted to the M1 (classically activated microglia phenotype) or M2 (alternatively activated microglia phenotype), as evidenced by an increased expression in iNOS (M1) or YM1 (M2). We further explored whether Notch1 is involved in regulating microglia/macrophage polarization. Immunofluorescence staining was performed to evaluate the expression of CD68, iNOS, and YM1 in the ischemic penumbra in both groups after MCAO (Figure 3A,B). The results showed that the number of CD68^+^iNOS^+^ cells was significantly reduced in the Notch1^M-KO^ group compared with the Notch1^FL/FL^ group (36.00 ± 3.41 vs. 68.80 ± 5.64, *p* < 0.001), and that the number of CD68^+^YM1^+^ cells notably increased in the Notch1^M-KO^ group compared with the Notch1^FL/FL^ group (56.60 ± 4.88 vs. 37.00 ± 3.90, *p* < 0.001) (Figure 3C,D). Western blot indicated that the expression of both the iNOS protein and the YM1 protein was statistically higher in the MCAO group than that in the sham group. This illustrated that there was an increase in the number of activated microglia infiltrating the ischemic area of the brain after CIRI. Notably, the expression level of the M1 marker protein iNOS significantly diminished in the Notch1^M-KO^ mice compared with the Notch1^FL/FL^ mice (0.65 ± 0.08 vs. 0.83 ± 0.07, *p* < 0.05), whereas the expression of the protein YM1 was elevated (0.75 ± 0.07 vs. 0.61 ± 0.09) (Figure 3E,F).

### 3.4. Notch1 Knockout Alleviates Pyroptosis Induced by CIRI

Cell pyroptosis plays an important role in the early pathophysiological stage of cerebral ischemia-reperfusion injury (CIRI). To study pyroptosis mechanism after CIRI, we used an animal model of MCAO in vivo. Western blot indicated that the expression of inflammasome constituents NLRP3 (0.76 ± 0.07 vs. 0.59 ± 0.08, *p* < 0.001) and Cl-caspase1 (0.84 ± 0.05 vs. 0.31 ± 0.06, *p* < 0.001) and members associated with pyroptosis Cl-GSDMD (0.78 ± 0.06 vs. 0.35 ± 0.06, *p* < 0.001), IL-1β (0.87 ± 0.08 vs. 0.53 ± 0.07, *p* < 0.001) and IL-18 (0.83 ± 0.08 vs. 0.55 ± 0.08, *p* < 0.001) was statistically and remarkably elevated in the MCAO group than in the sham group, confirming that after CIRI, the levels of the activated forms of the proinflammatory proteins IL-1β, IL-18 and caspase 1 were increased. Notably, myeloid-specific Notch1 deficiency effectively reduced the protein expression levels of members associated with pyroptosis and inflammasome constituents (Figure 4A,B). The gene expression levels of IL-1β, IL-18 and NLRP3 were increased in the MCAO group compared with the sham group according to qPCR validation. Further corroboration of the Western blot results, the gene expression levels of IL-1β (2.30 ± 0.31 vs. 3.57 ± 0.33, *p* < 0.001), IL-18 (2.30 ± 0.30 vs. 3.45 ± 0.42, *p* < 0.001) and NLRP3 (3.44 ± 0.44 vs. 4.80 ± 0.38, *p* < 0.001) were significantly lower in the Notch1^M-KO^ mice than in the Notch1^FL/FL^ mice (Figure 4C).

### 3.5. Microglial BV2 Cells Regulate OGD/R-Induced Apoptosis in HT22 Hippocampal Neurons via Notch1 In Vitro

To further investigate whether microglia affect neuronal apoptosis via the Notch1 pathway, we first established an in vitro cell co-culture system in which BV2 cells transfected with a lentivirus carrying NC-siRNA (control siRNA) or Notch1-siRNA (siRNA targeting Notch1) were plated in the upper chamber of a Transwell plate, and HT22 hippocampal neurons were plated in the lower chamber (Figure 5A). After the cells were co-cultured for 24 h, they were subjected to OGD/R, and apoptosis of HT22 neurons in the lower chamber after reoxygenation was measured in each group by flow cytometry. The experiment results of flow cytometry indicated that the proportion of HT22 apoptotic cells was remarkably increased in the OGD/R group compared with the control group, and the CCK-8 detection showed that the relative cell survival rate was diminished. Interestingly, the proportion of apoptotic cells after OGD/R was significantly lower for HT22 cells co-cultured with Notch1-siRNA-transfected BV2 cells than for HT22 cells co-cultured with NC-siRNA-transfected BV2 cells (0.31 ± 0.02 vs. 0.39 ± 0.04, *p* < 0.01) (Figure 5B,C), and the CCK-8 detection indicated that the relative cell survival rate was improved (0.78 ± 0.24 vs. 0.52 ± 0.13, *p* < 0.05) (Figure 5D). To further clarify the detailed mechanism of the effect of Notch1 in BV2 cells on neuronal apoptosis induced by OGD/R, Western blot was performed. The results revealed that the expression of the anti-apoptotic protein Bcl-2 (0.40 ± 0.06 vs. 0.65 ± 0.07, *p* < 0.01) was decreased, and the expression of the pro-apoptotic protein Bax (0.63 ± 0.07 vs. 0.32 ± 0.07, *p* < 0.001) significantly increased in HT22 cells that underwent OGD/R compared with the HT22 cells in control group. However, notably, after OGD/R, the Bcl2 protein expression level increased in HT22 cells co-cultured with Notch1-siRNA-transfected BV2 cells, compared with HT22 cells co-cultured with NC-siRNA-transfected BV2 cells (0.54 ± 0.05 vs. 0.40 ± 0.06, *p* < 0.01), while the expression of the protein Bax was obviously diminished (0.46 ± 0.05 vs. 0.63 ± 0.07, *p* < 0.01) (Figure 5E,F).

### 3.6. Notch1 Knockdown in Microglial BV2 Cells Inhibits OGD/R-Induced JAK2/STAT3 Activation and Pyroptosis in HT22 Cells

To further clarify the mechanism by which BV2 Notch1 regulates OGD/R-induced HT22 cell apoptosis, we performed Western blot analysis and found that the OGD/R group exhibited higher levels of inflammasome constituents NLRP3 (0.58 ± 0.03 vs. 0.37 ± 0.03, *p* < 0.001) and Cl-caspase1 (0.66 ± 0.09 vs. 0.36 ± 0.06, *p* < 0.001) and members associated with pyroptosis Cl-GSDMD (0.78 ± 0.08 vs. 0.33 ± 0.06, *p* < 0.001), IL-1β (0.73 ± 0.07 vs. 0.36 ± 0.05, *p* < 0.001) and IL18 (0.81 ± 0.06 vs. 0.31 ± 0.04, *p* < 0.001) than the control group. Notably, after OGD/R, the expression of inflammasome constituents NLRP3 (0.45 ± 0.06 vs. 0.58 ± 0.03, *p* < 0.05) and Cl-caspase1 (0.47 ± 0.07 vs. 0.66 ± 0.09, *p* < 0.01), pyroptosis-associated protein Cl-GSDMD (0.52 ± 0.06 vs. 0.78 ± 0.08, *p* < 0.001), IL-1β (0.53 ± 0.09 vs. 0.73 ± 0.07, *p* < 0.01) and IL18 (0.54 ± 0.08 vs. 0.68 ± 0.06, *p* < 0.01) was statistically and sensibly lower in the HT22 cells co-cultured with Notch1-siRNA-transfected BV2 cells than in the HT22 cells co-cultured with NC-siRNA-transfected BV2 cells (Figure 6A,B). Comparing with the control group, qPCR analysis of inflammasome constituents (NLRP3 and Cl-caspase1) and members associated with pyroptosis (Cl-GSDMD, IL-1β and IL-18) verified that their gene expression was upregulated in the OGD/R group. Consistent with the Western blot results, qPCR revealed that the expression levels of pyroptosis-associated genes were reduced in HT22 cells co-cultured with Notch1-siRNA-transfected BV2 cells compared with HT22 cells co-cultured with NC-siRNA-transfected BV2 cells after OGD/R (Figure 6C). To investigate the molecular mechanism through which Notch1 in BV2 cells affects pyroptosis in HT22 neurons, we utilized the STRING database and Cytoscape software to construct the PPI network of Notch1, and the results indicated an interaction between Notch1 and JAK2/STAT3 (Figure 6D). The results of Western blot analysis indicated that the P-JAK2/JAK2 and P-STAT3/STAT3 ratios were obviously increased in the OGD/R group compared with the control group. Interestingly, after OGD/R, the P-JAK2/JAK2 and P-STAT3/STAT3 ratios in HT22 cells co-cultured with Notch1-siRNA-transfected BV2 cells were markedly diminished compared to those in HT22 cells co-cultured with the NC-siRNA-transfected BV2 cells (Figure 6E,F).

## 4. Discussion

After cerebral ischemic injury, Notch1 signaling is significantly elevated in mice [21]. Studies have shown that in the pathophysiological mechanism of cerebral ischemia, Notch1 promotes the process of pyroptosis of neurons, in addition to activating microglia and promoting the cerebral infiltration of lymphocytes [22]. Consistent with this conclusion, we found that in the acute phase in cerebral ischemia, the expression of Notch1 in the ischemic penumbra was upregulated, and this change was accompanied by extensive activation of microglia. In addition, we found that microglia can regulate neuronal pyroptosis induced by anoxia/reoxygenation through Notch1. Compared with Notch1^FL/FL^ mice, Notch1^M-KO^ mice showed a reduced infarct volume and improved neurological function scores during the acute phase of ischemia reperfusion injury. Those changes are related to the suppression of cell pyroptosis. Evidence suggests that the inflammatory response cascade has a significant impact on the pathophysiological process of cerebral ischemia injury. Leukocyte infiltrating from the periphery into the cerebral ischemic region activates microglia, and the activated microglia further increases the expression of inflammatory factors, which in turn exacerbates the brain injury [23]. Therefore, the regulation of inflammatory immune cascade response in response to cerebral infiltration of peripheral immune cells has gradually become a hot topic of recent research, and has also become the most discussed new therapeutic target for developing targeted inflammatory regulatory hubs. 

However, cerebral ischemia–reperfusion injury is prone to involve more than just an inflammatory response. Pyroptosis is also considered to be the pivotal link in the regulation of neurological function injury after cerebral ischemia. Pyroptosis distinct from necrosis or apoptosis is a common form of proinflammatory programmed cell death. It is characterized by the rapid disintegration of the cell membrane and release of pro-inflammatory mediators and inflammatory cytokines from the cellular contents [14]. Typically, pyroptosis is closely related to NLRP3. NLRP3 inflammasomes activated after brain I/R injury are first formed in microglia and are expressed predominantly in neurons after 24 h [24]. The inflammatory vesicle NLRP3 serves as a molecular platform to activate caspase-1, which acts as an effector to cleave proteins and process interleukinogen IL-1β and IL-18 into mature forms, which are secreted into the extracellular space to exercise the next step in the inflammatory response cascade [10]. In addition, activated caspase-1 cleaves GSDMD and triggers oligomerization of the intracellular GSDMD-N structural domain, resulting in the formation of pores for the release of IL-1β, IL-18.

In this study, we showed that the expression of inflammasome components (Cl-caspase1 and NLRP3) was increased after MCAO in mice. In contrast, myeloid-specific Notch1 knockout effectively inhibited the expression of protein Cl-GSDMD, IL1β and IL-18 (pyroptosis-related proteins) after MCAO. Therefore, we speculate that the Notch1 gene in myeloid cells plays a critical role in regulating cell pyroptosis after cerebral ischemia in mice. Additionally, through the co-culture of microglia and HT22 cells in vitro, it was further verified that specific knockdown of Notch1 in microglia suppressed OGD/R-induced neuronal pyroptosis. Consistent with this result, specific knockdown of Notch1 in microglia restrained neuronal apoptosis. The expression of Bcl2 was upregulated and apoptosis related proteins Bax was effectively depressed in HT22 cells co-cultured with Notch1-siRNA transfected BV2 cells. This research differs from previous studies in that we connect the inflammatory immune cascade with the cell pyroptosis network. The occurrence of inflammatory response and cell pyroptosis is not independent, and there may be some connection between them. This phenomenon is noteworthy, and there may be mutual regulation between inflammatory response and cell pyroptosis. Our findings provide important preliminary clues for further exploration of a clear regulatory mechanism between the two in the future.

Based on previous research and the results of this experiment, we propose that myeloid-specific Notch1 knockout causes the polarization direction of cerebral microglia/macrophages to lean towards M2 (alternatively activated microglia phenotype), releasing anti-inflammatory cytokines, inhibit the activation of NLRP3 inflammasomes in neurons, and inhibit the cleavage and activation of caspase-1 precursor to produce caspase-1. The lysis of Gasdermin D (GSDMD) is inhibited, thereby preventing the recruitment of the N domain of GSDMD to the cell membrane, leading to the formation of membrane pores and ultimately inhibiting neuronal pyroptosis. The Janus kinase (JAK) and signal transducer and activator of transcription (STAT) pathways regulate dozens of cellular processes, such as cell proliferation, differentiation, migration, apoptosis and survival [25]. STAT3 can directly transmit signals to the nucleus and increase the expression of genes encoding proteins involved in neuroinflammation [26]. Our in vitro studies also showed that after OGD/R, the protein expression of P-JAK2 and P-STAT3 in HT22 cells co-cultured with Notch1-siRNA transfected BV2 cells was significantly reduced compared to HT22 cells co-cultured with NC-siRNA transfected BV2 cells. Those results suggest that Notch1 knockdown in microglia inhibited anoxia/reoxygenation-induced JAK2/STAT3 activation and pyroptosis in hippocampal neuron HT22 cells. Previous research has shown that pharmacological inhibition of the JAK2/STAT3 pathway can reduce inflammatory response after cerebral ischemia and exert neuroprotective effects [19]. In addition, studies have shown that Notch1 signaling prevents burn-induced cardiac injury by regulating JAK2/STAT3 [27]. However, there has been no study exploring whether JAK2/STAT3 is directly associated with cell pyroptosis. Our research findings provide important preliminary clues for further exploring the specific mechanism by which JAK2/STAT3 affects the process of cell pyroptosis.

There are some limitations in this study. Firstly, in cerebral ischemia-reperfusion injury, pyroptosis occurs in various cells (such as astrocytes, brain-resident microglia, monocyte derived macrophage, neurons, etc.), producing cytokines and inflammasomes. Recent studies have shown that inflammasomes activated after brain I/R injury first generate in microglia. Then, the inflammasomes are mainly expressed in neurons and vascular endothelial cells after 24 h, mainly in neurons [24]. In this study, we focused on the pyroptosis of neurons. In the future, we will further investigate the specific mechanisms of pyroptosis in other cells during CIRI. Secondly, although our data suggested that JAK2/STAT3 pathway was involved in microglia Notch1-mediated pyroptosis of neurons, further work is necessary to confirm whether JAK2/STAT3 axis is essential for Notch1-mediated pyroptosis. Finally, further studies are needed in the future to investigate the specific mechanisms of cell pyroptosis during CIRI in affecting neurological function in ischemic stroke. In addition, the inter-crosstalk between the inflammatory immune cascade response and cell pyroptosis needs to be further explored in the future to look for internal liaisons among them in order to search for specific mechanisms by which new peripheral myeloid immune cell-specific targets modulate the balance of immune-inflammatory responses in the central nervous system.

## 5. Conclusions

In summary, the above results indicate that myeloid-specific Notch1 knockout has a beneficial effect on the neuroprotection of CIRI, partly by polarizing microglia towards M2, inhibiting neuronal pyroptosis and JAK/STAT3 pathway activation, thereby protecting neurons from damage induced by anoxia/reoxygenation. Our findings combine the specific mechanisms of immune-inflammatory cascade response and neuronal pyroptosis in ischemic stroke, providing important research clues for further in-depth investigation of the liaison mechanism in the future. Additionally, regulating the expression of Notch1 in myeloid cells has become a potential therapeutic target for future development of improving the prognosis of ischemic stroke disease. It has an important clinical application value for improving the treatment outcome of ischemic stroke patients in the future.

## Figures and Tables

**Figure 1 brainsci-13-01657-f001:**
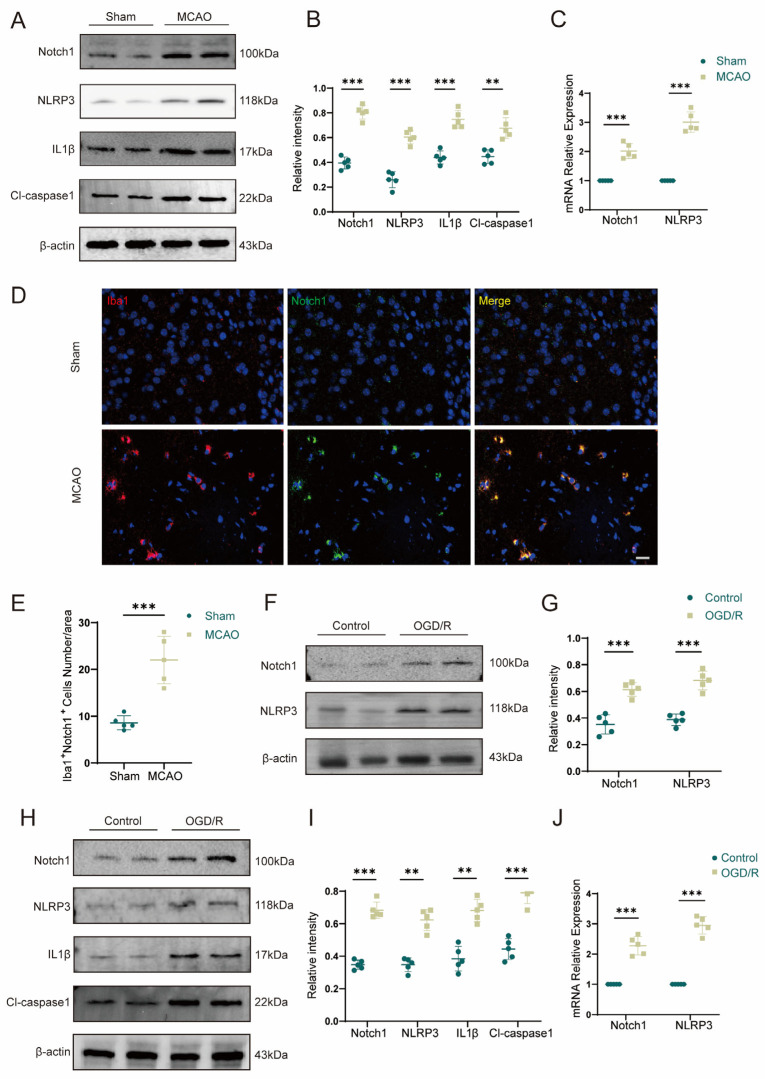
Notch1 signaling is closely associated with ischemia–reperfusion injury in the mouse brain. (**A**) Representative Western blot images of Notch1, NLRP3, IL1β, and Cl-caspase1 expression in the ischemic penumbra of wild-type mice 3 days after MCAO or sham surgery (n = 5 mice/group). (**B**) Quantification of the relative protein expression of Notch1, NLRP3, IL1β, and Cl-caspase1 in the ischemic penumbra in the MCAO group compared with the sham-operated group. (**C**) The effects of MCAO and sham operation on Notch1 and NLRP3 mRNA levels in the ischemic penumbra of wild-type mice 3 days after the operation. (**D**) Representative images of immunofluorescence staining with antibodies against Notch1 (green) and Iba1 (red). Cell nuclei were labeled with DAPI (blue). n = 5 mice/group. (**E**) The number of Iba1^+^Notch1^+^ cells. (**F**) Representative Western blot images Notch1 and NLRP3 expression in BV2 cells subjected to OGD/R and control group. (**G**) Quantification of the relative protein expression of Notch1 and NLRP3 in BV2 cells subjected to OGD/R and control group. (**H**) Representative Western blot images Notch1, NLRP3, IL1β and Cl-caspase1 expression in HT22 cells subjected to OGD/R and control group. (**I**) Quantification of the relative protein expression of Notch1, NLRP3, IL1β and Cl-caspase1 in HT22 cells subjected to OGD/R and control group. (**J**) mRNA levels of Notch1 and NLRP3 in HT22 cells subjected to OGD/R and control HT22 cells. Mean ± SD. n = 5 in each group. ** *p* < 0.01, *** *p* < 0.001. Scale bar, 20 μm.

**Figure 2 brainsci-13-01657-f002:**
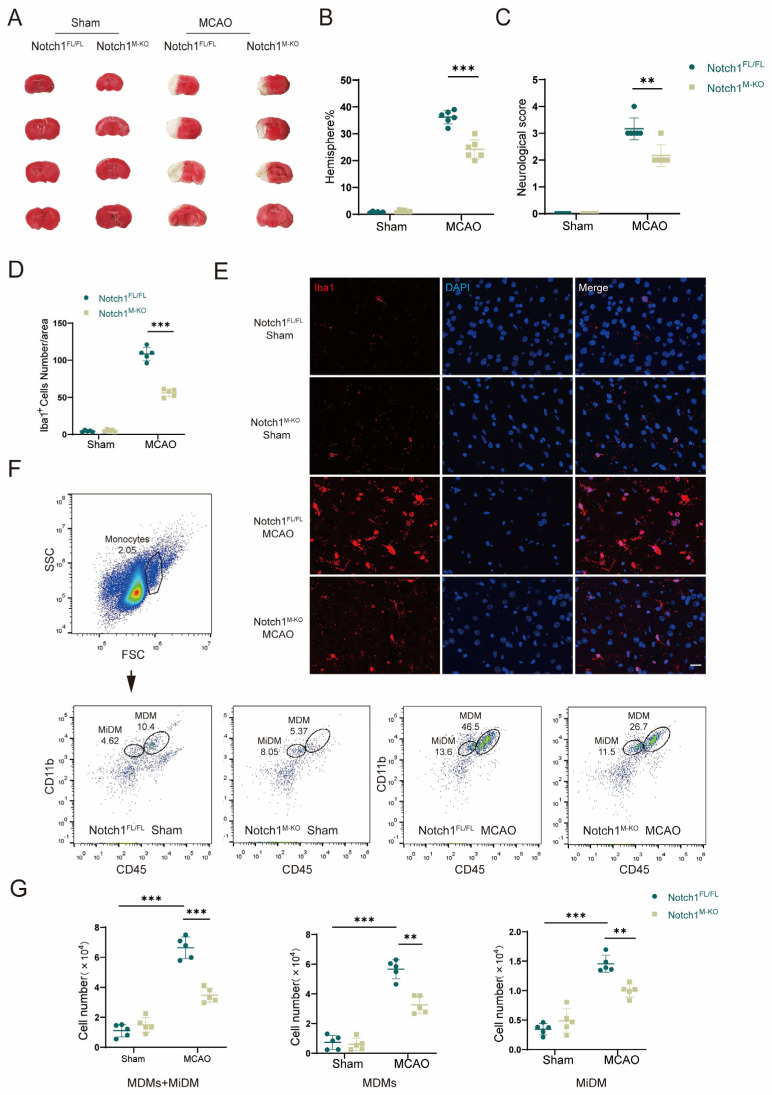
Notch1 knockout attenuates CIRI and cerebral migration of monocyte-derived macrophages. (**A**) TTC staining for cerebral infarct analysis. (**B**) Statistical analysis of the infarct volume according to TTC staining. The infarct size is expressed as percentage of infarct volume to contralateral hemisphere size 3 days after MCAO (n = 6 mice/group). (**C**) Statistical analysis of neurological function scores 3 days after MCAO (n = 6 mice/group). (**D**) Statistical analysis of the number of Iba1^+^ cells. (**E**) Representative images of immunofluorescence staining with an antibody against Iba1 (red). Cell nuclei were labeled with DAPI (blue). (n = 5 mice/group) (**F**) Gating strategy to identify CD45^inter^CD11b^+^ cells (MiDM) and CD45^high^CD11b^+^ cells (MDMs). (**G**) Statistical analyses of the numbers of MiDM, MDMs and total monocytes. (n = 5 mice/group) Mean ± SD. ** *p* < 0.01, *** *p* < 0.001. Scale bar, 20 μm.

**Figure 3 brainsci-13-01657-f003:**
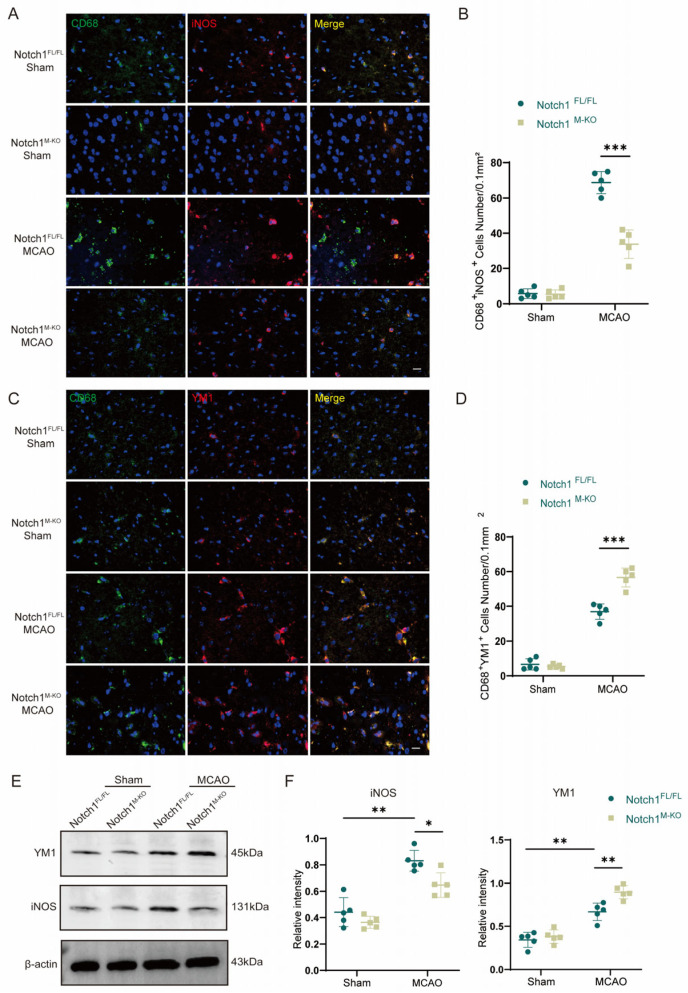
Notch1 knockout promotes M2 phenotypes in the ischemic penumbra. (**A**) Representative images of immunofluorescence staining with antibodies against CD68 (green) and iNOS (red). (**B**) Statistical analysis of the number of CD68^+^iNOS^+^ cells. (**C**) Representative images of immunofluorescence staining with antibodies against CD68 (green) and YM1 (red). Cell nuclei were labeled with DAPI (blue). (**D**) Statistical analysis of the number of CD68^+^YM1^+^ cells. (**E**) Representative Western blot bands and (**F**) quantitative analysis; target protein levels were normalized to β-actin levels. Mean ± SD. n = 5 in each group. * *p* < 0.05, ** *p* < 0.01, *** *p* < 0.001. Scale bar, 20 μm.

**Figure 4 brainsci-13-01657-f004:**
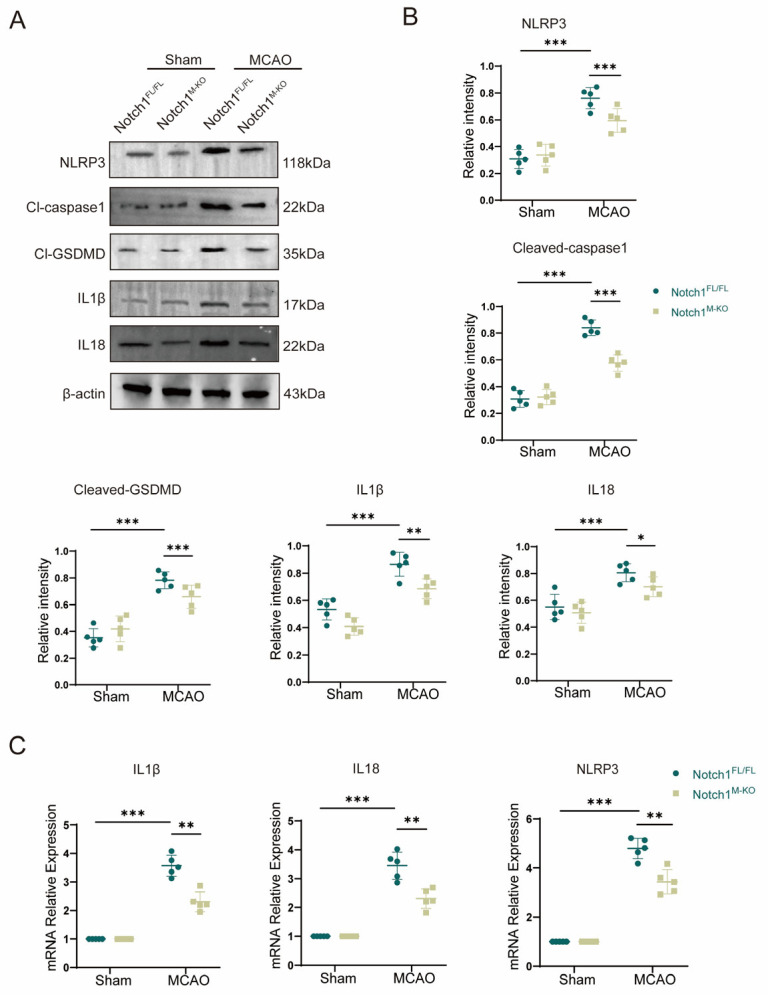
Notch1 knockout alleviates CIRI-induced pyroptosis. (**A**) Representative Western blot bands and (**B**) Quantitative analysis; target protein levels were normalized to β-actin levels. (**C**) mRNA levels of inflammasome components (NLRP3) and pyroptosis-related proteins (IL-1β and IL-18) in Notch1^FL/FL^ and Notch1^M-KO^ mice in the MCAO group and sham group. Mean ± SD. n = 5 in each group. * *p* < 0.05, ** *p* < 0.01, *** *p* < 0.001.

**Figure 5 brainsci-13-01657-f005:**
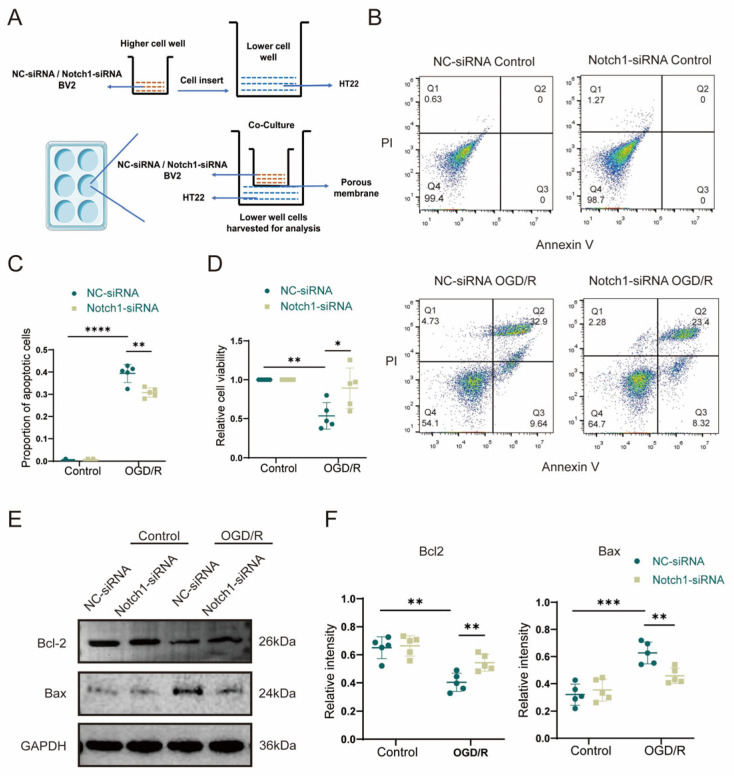
BV2 microglia regulate OGD/R-induced apoptosis in HT22 cells via Notch1 in vitro. (**A**) Schematic diagram of the BV2 cell and HT22 cell co-culture system. BV2 cells stably transfected with NC-siRNA lentivirus or Notch1-siRNA lentivirus were plated in the upper chamber, and HT22 hippocampal neurons were plated in the lower chamber. After 24 h, the co-cultured cells were subjected to OGD/R, and HT22 cells in the lower chamber were collected 12 h after reoxygenation for analysis. (**B**) Flow cytometry analysis of apoptosis. (**C**) Statistical analysis of the percentage of apoptotic cells. (**D**) Analysis of relative cell viability by the CCK-8 assay. (**E**) Western blot analysis of Bcl2 and Bax protein expression. (**F**) Relative quantification of Bcl2 and Bax protein expression. Mean ± SD. n = 5 in each group. * *p* < 0.05, ** *p* < 0.01, *** *p* < 0.001, **** *p* < 0.0001.

**Figure 6 brainsci-13-01657-f006:**
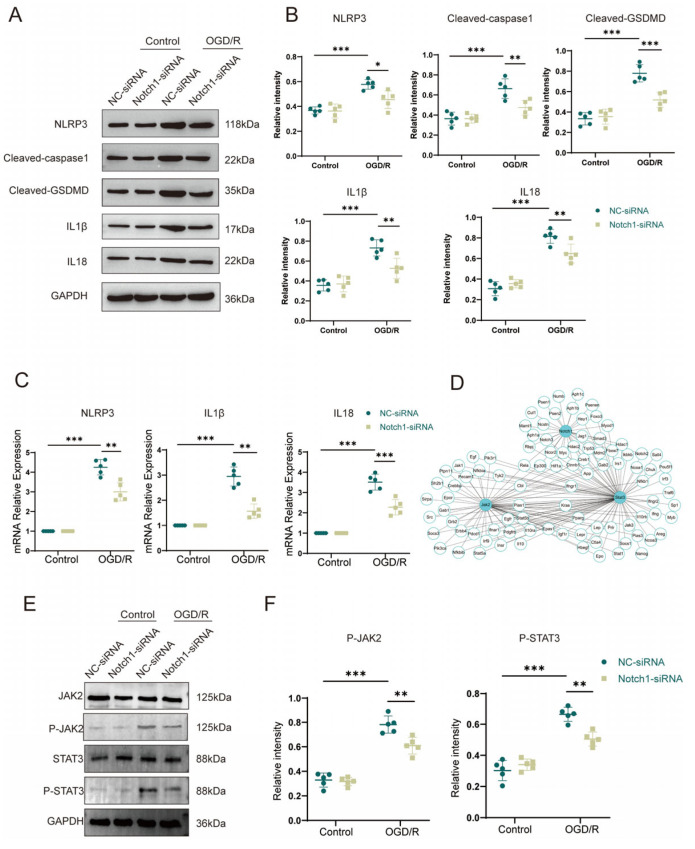
Notch1 knockdown in microglial BV2 cells inhibits OGD/R-induced JAK2/STAT3 activation and pyroptosis in HT22 cells. (**A**) Western blot analysis of the expression of pyroptosis-associated proteins in HT22 cells co-cultured with NC-siRNA-transfected BV2 cells and HT22 cells co-cultured with Notch1-siRNA-transfected BV2 cells. (**B**) Relative quantification of pyroptosis-associated protein expression in HT22 cells. (**C**) mRNA expression levels of inflammasome-associated proteins (NLRP3 and Cl-caspase1) and pyroptosis-associated proteins (Cl-GSDMD, IL-1β and IL-18) in HT22 cells co-cultured with NC-siRNA-transfected BV2 cells and those co-cultured with Notch1-siRNA-transfected BV2 cells after OGD/R or sham treatment. (**D**) Protein–protein interaction (PPI) network of Notch1 and JAK2/STAT3. (**E**) Representative Western blot images of P-JAK2, JAK2, P-STAT3, and STAT3 in HT22 cells co-cultured with NC-siRNA-transfected BV2 cells and HT22 cells co-cultured with Notch1-siRNA-transfected BV2 cells after OGD/R or sham treatment. (**F**) Relative quantification of P-JAK2 and P-STAT3 protein expression. Mean ± SD. n = 5 in each group. * *p* < 0.05, ** *p* < 0.01, *** *p* < 0.001.

**Table 1 brainsci-13-01657-t001:** Primers for RT–PCR.

Genes		Primers (5’–3’)
IL1β	F	CACCTCTCAAGCAGAGCACAG
	R	GGGTTCCATGGTGAAGTCAAC
IL18	F	CAATGGTTCCTTCATTGAGC
	R	AACAAACAGGAGAAGTTGGT
NLRP3	F	GTGGTGACCCTCTGTGAGGT
	R	TCTTCCTGGAGCGCTTCTAA
Notch1	F	GATGGCCTCAATGGGTACAAG
	R	TCGTTGTTGTTGATGTCACAGT
GAPDH	F	AGGTCGGTGTGAACGGATTTG
	R	TGTAGACCATGTAGTTGAGGTCA

## Data Availability

Data are contained within the article and Appendix A.

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
