# Peer review of "Targeting Microglia/Macrophages Notch1 Protects Neurons from Pyroptosis in Ischemic Stroke"

_brainsci, 2023, doi:10.3390/brainsci13121657_

Round 1

Reviewer 1 Report

Comments and Suggestions for Authors

The study performed by authors “Targeting Microglia/Macrophages Notch1 Protects Neurons from Pyroptosis in Ischemic Stroke”, This work showed some interesting information. there are some minor points to be further addressed. Mostly linked to the research topic and experimental results. However, the manuscript needs major revision and following questions/points should be further clarified

Some questions and comments as follow:

 Abstract:

1. The abstract mentions that myeloid Notch1 deficiency decreased cerebral infarct volume, neurological function scores, and infiltration of peripheral monocytes/macrophages, but it does not provide the actual numeric values or statistical significance of these findings. Including quantitative results would make the abstract more informative and credible.

2.  The abstract briefly mentions that targeting myeloid-specific Notch1 is an effective strategy for the treatment of ischemic stroke, but it does not elaborate on the potential implications or significance of these findings. Providing a brief discussion on the potential therapeutic implications of the study would make the abstract more impactful.

Overall, improving the abstract by providing more context, specific details on methods and results, discussing the implications, and adding a clear conclusion would make it more informative and engaging for readers

Discussion:

3.  the discussion does not adequately connect the findings of the study with previous research in the field. While some references are mentioned, there is little analysis or comparison with existing literature, limiting the contextual understanding of the study's contribution.

4. The discussion lacks in-depth interpretation of the results and their implications. It primarily focuses on describing the experimental findings without discussing their significance or providing insights into the underlying mechanisms. A more detailed analysis and interpretation of the results would strengthen the discussion.

5.The discussion briefly mentions limitations, such as the involvement of various cell types in pyroptosis and the necessity of further investigations on the JAK2/STAT3 axis. However, these limitations are not thoroughly discussed or acknowledged in terms of their impact on the study's findings or generalizability. Providing a comprehensive discussion of the limitations would improve the transparency and credibility of the study.

6.  The discussion does not provide clear suggestions or avenues for future research based on the study's findings. Identifying potential areas for further investigation and expanding upon the implications of the results would enhance the scientific value and practical applications of the study.

Author Response

Revierwer1

We sincerely appreciate your revision suggestions. We have made corrections in the revised manuscript. Meanwhile, the manuscript has been reviewed and edited by the language services department of the American Journal of Experts (AJE). We have underlined all the revisions in the latest uploaded version of manuscript. In addition, we would like to provide a detailed explanation as follows:

  1. The abstract mentions that myeloid Notch1 deficiency decreased cerebral infarct volume, neurological function scores, and infiltration of peripheral monocytes/macrophages, but it does not provide the actual numeric values or statistical significance of these findings. Including quantitative results would make the abstract more informative and credible.

Response:

As suggested, we have added the statistical information in the Results and Abstract section, please refer to our revised manuscript.

  1. The abstract briefly mentions that targeting myeloid-specific Notch1 is an effective strategy for the treatment of ischemic stroke, but it does not elaborate on the potential implications or significance of these findings. Providing a brief discussion on the potential therapeutic implications of the study would make the abstract more impactful.

Response:

We are very grateful for your suggestions. In order to further improve the clinical application value of the findings of this study, we have added relevant research significance in the Abstract conclusion section. Please refer to our revised manuscript for details. Our findings elucidate the underlying mechanism of the myeloid Notch1 signaling pathway in regulating neuronal pyroptosis in CIRI, suggesting that targeting myeloid-specific Notch1 is an effective strategy for the treatment of ischemic stroke. The development of this target has clear preclinical value for improving the outcome and long-term prognosis of ischemic stroke patients in the future.

  1. The discussion does not adequately connect the findings of the study with previous research in the field. While some references are mentioned, there is little analysis or comparison with existing literature, limiting the contextual understanding of the study's contribution.

Response:

Thanks for your helpful comments. We have added the related information in the Discussion section (Line 479-484, Line 492-496, Line 521-527, Line 572-578).

  1. The discussion lacks in-depth interpretation of the results and their implications. It primarily focuses on describing the experimental findings without discussing their significance or providing insights into the underlying mechanisms. A more detailed analysis and interpretation of the results would strengthen the discussion.

Response:

Thank you for the valuable suggestions. We have modified the Discussion section. We have added the exploration of the underlying mechanisms based on the findings and presented the pre-foundational implications of this study for further in-depth research in the future (Line 521-527, Line 572-578).

  1. The discussion briefly mentions limitations, such as the involvement of various cell types in pyroptosis and the necessity of further investigations on the JAK2/STAT3 axis. However, these limitations are not thoroughly discussed or acknowledged in terms of their impact on the study's findings or generalizability. Providing a comprehensive discussion of the limitations would improve the transparency and credibility of the study.

Response:

We would greatly appreciate your suggestions. This study has certain limitations. Firstly, in cerebral ischemia-reperfusion injury, pyroptosis occurs in various cells (such as astrocytes, brain-resident microglia, monocyte derived macrophage, neurons, etc.), producing cytokines and inflammasomes. Recent studies have shown that inflammasomes activated after brain I/R injury first generate in microglia. Then the inflammasomes are mainly expressed in neurons and vascular endothelial cells after 24 h, mainly in neurons. In this study, we focused on the pyroptosis of neurons. In the future, we will further investigate the specific mechanisms of pyroptosis in other cells during CIRI. Secondly, although our data suggested that JAK2/STAT3 pathway was involved in microglia Notch1-mediated pyroptosis of neurons, further work is necessary to confirm whether JAK2/STAT3 axis is essential for Notch1-mediated pyroptosis. Finally, further studies are needed in the future to investigate the specific mechanisms of cells pyroptosis during CIRI in affecting neurological function in ischemic stroke. In addition, the inter-crosstalk be-tween the inflammatory immune cascade response and cells pyroptosis needs to be fur-ther explored in the future to look for internal liaisons among them in order to search for specific mechanisms by which new peripheral myeloid immune cell-specific targets modulate the balance of immune-inflammatory responses in the central nervous system.

Based on the existing research conclusions and initial research design, we have summarized the relevant shortcomings and proposed relevant directions for sustainable research based on existing conclusions, providing strategic research value for future development in this field. Our team will further deepen our exploration in this field, hoping to clarify the underlying mysterious mechanism between inflammatory immune response and other cells pyroptosis in ischemic stroke in the future. For the revised content, please refer to the Discussion section of the latest uploaded manuscript (Line 552-568).

  1. The discussion does not provide clear suggestions or avenues for future research based on the study's findings. Identifying potential areas for further investigation and expanding upon the implications of the results would enhance the scientific value and practical applications of the study.

Response:

Thanks for your suggestions. We further clarify the research value and significance of this study in the Discussion section, and objectively propose the contribution of research conclusions to the future development of ischemic stroke related fields. Please refer to our latest submitted manuscript for the modifications (Line 572-578).

Reviewer 2 Report

Comments and Suggestions for Authors

In the manuscript entitled “Targeting Microglia/Macrophages Notch1 Protects Neurons from Pyroptosis in Ischemic Stroke” the authors look at the effect of Notch1 on neuroprotection following cell death by pyroptosis in the context of cerebral ischemia. These are the main concerns about this manuscript:

1.         Is stroke really the main cause of death worldwide?

2.  The introduction should make a clear distinction between inflammasome activation and pyroptosis. Caspase-1 can indeed cleave GSDM-D to induce pyroptosis. The cell lysis that causes the release of LDH in pyroptosis is mediated by ninjurin1. Not sure why cystatinase-1 is abbreviated as (GSDMD), and why the authors state that it is responsible for perforating the cell membrane. GSDM-D-N does make a pore to induce release of IL-1beta but that is not clear in the introduction. Please, correct the introduction.

3.         In 2.7, the actual details of the flow cytometry procedure are missing. As of now it mostly describes the cell isolation portion of the experiment.

4.         In 2.11, what media was used?

5.         In 2.11, what is the appropriate time that the authors are referring to? The length of the reperfusion period does seem excessive.

6.   What post-hoc tests were used? Were all the data normally distributed?

7.         The ICC quality in all the images is too small to be able to discern the findings of the study.

8.         In figure 1, add caspase-1 and IL-1beta to the western blot analysis.

9.         Figure 2A needs to be made bigger.

10.     The M1/M2 concept in microglia is outdated. The manuscript needs to be rephrased to observe the current understanding on microglia phenotype.

11.     In 4B, the quantification of NLRP3 does not reflect what is shown in the representative gel in 4A. Neither for caspase-1, GSDM-D and IL-18. Same thing for all proteins in 6A and B.

12.     What is the rationale for looking into apoptosis?

13.     Bcl2 quantification in figure 5 does not match the representative image of the gel.

14.     Figure 6D is very hard to read.

15.     In 6E, P-JAK2 data does not look significant.

16.     In the discussion, the M1/M2 discussion should be avoided.

17.     The discussion needs a more thorough description of what has been already shown about the inflammasome and stroke.

Author Response

Reviewer2

We sincerely appreciate your revision suggestions. We have made corrections in the revised manuscript. Meanwhile, the manuscript has been reviewed and edited by the language services department of the American Journal of Experts (AJE). We have underlined all the revisions in the latest uploaded version of manuscript. In addition, we would like to provide a detailed explanation as follows:

  1. Is stroke really the main cause of death worldwide?

Response:

We have reviewed a large number of epidemiological and clinical studies. Stroke is a common disease, with one in four people affected over their lifetime, and is the second leading cause of death and third leading cause of disability in adults worldwide1. Using Global Burden of Disease Study estimates of stroke incidence and the competing risks of non-stroke mortality, the study published on N Engl J Med estimated the cumulative lifetime risk of ischemic stroke, hemorrhagic stroke, and total stroke (with 95% uncertainty intervals [UI]) for 195 countries among adults over 25 years) for the years 1990 and 2016 and according to the GBD Study Socio-Demographic Index (SDI). Stroke accounts for almost 5% of all disability-adjusted life years (DALYs) and 10% of all deaths worldwide, with the bulk of this burden (over 75% of deaths from stroke and 81% of DALYs) falling on low- and middle-income countries. The total global burden of stroke is increasing2. Synthesizing the existing research context, stroke is the main cause of death and physical disability worldwide. We hope that our answers have addressed your concerns.

  1. The introduction should make a clear distinction between inflammasome activation and pyroptosis. Caspase-1 can indeed cleave GSDM-D to induce pyroptosis. The cell lysis that causes the release of LDH in pyroptosis is mediated by ninjurin1. Not sure why cystatinase-1 is abbreviated as (GSDMD), and why the authors state that it is responsible for perforating the cell membrane. GSDM-D-N does make a pore to induce release of IL-1beta but that is not clear in the introduction. Please, correct the introduction.

Response:

Thank you for your suggestions. We have made a revision in the Introduction section. Please refer to our revised manuscript (Line 62-67). Caspase-1 is activated by various inflammasome complexes in innate immunity3. The primary mechanism of cells pyroptosis is hypothesized to be NLRP3-inflammatory vesicles of immune cells in vivo that are activated by pathogens. It further induces the local aggregation of inactive cysteine precaspase-1 (Pro-caspase-1) and promotes its hydrolysis to active cysteine activity-1 (Caspase-1), which shears inactive IL-1β precursor and IL-18 precursor into active IL-1β and IL-18, causing cellular pyroptosis4-6. The Gasdermin-N domain of GSDMD forms membrane pores to trigger pyroptosis. In other words, cleaved gasdermin D (Cl-GSDMD) or GSDMD-N perforates the cell membrane, disintegrating it and releasing inflammatory cellular contents, and mature inflammatory cytokines (IL1β, etc.) inducing an inflammatory response cascade or initiating a cell death program.

  1. In 2.7, the actual details of the flow cytometry procedure are missing. As of now it mostly describes the cell isolation portion of the experiment.

Response:

Thanks for your suggestions. In order to clarify the specific operational details related to flow cytometry, we have supplemented the procedure. Please refer to paragraph 2.7 (Line 185-199) of the latest submitted manuscript Materials and Methods for details. After centrifugation, a white flocculent layer (single nucleated cells, including lymphocytes) was observed at the delamination of the 70% and 37% Percoll solutions; the white flocculent layer was collected with a pipette and centrifuged at 1200 r/min for 10 min. Cells were washed with FACS buffer and incubated with CD45 (553079, BD Bioscience) and CD11b antibodies (550993, BD Bioscience) (protected from light and frozen for 30 min). CytoFLEX flow cytometer (Beckman Coulter) was used for analysis. The fluorescence channels corresponding to CD45 (FITC) and CD11b (PerCP-Cy5.5) were selected. A new FSC-SSC scatter plot was created, giving each sample a settled name (in-cluding the blank tube). The X-axis and Y-axis maximum settings were adjusted so that the cells were in the appropriate position in the center of the scatter plot. Each gate of the target cells group was circled and applied to the histogram. The compensation was ad-justed appropriately. About 100,000 cells were collected in each group for analysis on CytoFLEX flow cytometer (Beckman Coulter). The expression of target molecules was analyzed using (v2.3, Beckman Coulter) and the final results were processed using FlowJo software.

  1. In 2.11, what media was used?

Response:

We have added the information about the media related to the cell co-culture experiments in paragraph 2.11 of the manuscript (Line 238-242). In control group, we used high-glucose medium (C11995500BT, Dulbecco's Modified Eagle Medium, Gibco, China), with 10% FBS (11011-8611, TianHang, Biotechnology Co., Ltd, China). In OGD/R group, we used glucose-free medium (PM150270, Dulbecco's Modified Eagle Medium, Procell Life Sci-ence&Technology Co.,Ltd, China).

  1. In 2.11, what is the appropriate time that the authors are referring to? The length of the reperfusion period does seem excessive.

Response:

Thanks for your questions. We have consulted a large number of references for the questions raised by the reviewers. Based on the previous research, we chose to induce cell OGD/R in glucose-free DMEM culture medium for 6 hours7 in a hypoxic chamber and then perfused for 12 hours8 to correspond to the acute cerebral ischemic event of the in vivo animal experiments, which is also known as the MCAO model. In order to show the details of our methodology more clearly, we have added the exact timing of hypoxia modeling explicitly in the revised manuscript (Line 247-254).

  1. What post-hoc tests were used? Were all the data normally distributed?

Response:

Thanks for your advice. Our data satisfy the following statistical requirements: 1. the sample data conform to a normal distribution; 2. the sample data satisfy the requirement of chi-square; and 3. the data are independent of each other. Based on this, the data were analyzed by Student’s t test for two group comparisons or one-way analysis of variance (ANOVA), followed by Dunnett’s post hoc test for multiple comparisons. We have described the statistical methods in our revised manuscript (Line 276-280).

  1. The ICC quality in all the images is too small to be able to discern the findings of the study.

Response:

As suggested, we have enlarged the representative images of immunofluorescence in the revised version (Figure 1,2,3).

  1. In figure 1, add caspase-1 and IL-1beta to the western blot analysis.

Response:

As suggested, we have uploaded the latest version of Figure 1.

  1. Figure 2A needs to be made bigger.

Response:

Thanks for your suggestion. We have enlarged the image in the revised version (Figure 2A).

  1. The M1/M2 concept in microglia is outdated. The manuscript needs to be rephrased to observe the current understanding on microglia phenotype.

Response:

Regarding the inaccuracy of the M1/M2 concept raised by the reviewers, we acknowledge the limitations of the typing method for microglia in this study. According to the latest findings, there has been no consensus on the topic of microglial cell polarization, and the subtypes of microglia may coexist in the CNS and microglia show different characteristics in different environments9. Our understanding of the concept of microglia typing is insufficient, and it is difficult to proceed to more systematic and in-depth studies without appropriate microglia typing methods.

Microglia and macrophages are rapidly activated after ischemia, and they are important members of the immune cascade after acute ischemic events. Activated macrophages are converted to the M1 (classically activated microglia phenotype) or M2 (alternatively activated microglia phenotype). With appropriate stimulation, M1 microglia are the first line of defense for the neuroimmune system, and they usually appear within a few hours of stimulation. M2 microglia are protective cells that secrete anti-inflammatory factors and upregulate neuroprotective factors. They play an active role in immune defense by providing protection against organisms, fighting infection, acting as antigen-presenting cells, killing tumor cells, performing tissue repair, providing chemotactic targeting, and eliminating invading bacteria.

M1/M2 microglia typing cannot show the complete biological function of microglia. Therefore, we need a new nomenclature system to redefine microglia and their subtypes. If the M1/M2 phenotypic is no longer used, this also implies that there are limitations in determining the activation status of microglia based only on the typical markers of M1/M2. However, as of now, no newer and more rational way of outlining a reasonable expression of both pro-inflammatory and pro-repair phenotypes of microglia has been found, and therefore we have continued to follow the typing representation that is still in use9, 10. We are confident that in future studies we will be able to identify a more rational expression of microglia phenotypes that can accurately characterize their different functions and subtype transformations, and in the future, we will further improve on the limitations of this study. Even so, we are very receptive to the reviewers' suggestions and are willing to apply them to our article (Line 352-357). We hope that our explanations and modifications will satisfy the reviewers.

  1. In 4B, the quantification of NLRP3 does not reflect what is shown in the representative gel in 4A. Neither for caspase-1, GSDM-D and IL-18. Same thing for all proteins in 6A and B.

Response:

We would like to express our sincere gratitude to the reviewers for their suggested revisions. In order to present the results of Western blot experiments more clearly, we have re-selected more representative protein bands. Please refer to our newly submitted manuscript Figure 4A, B and Figure 6A, B. We hope that the figures we re-uploaded can provide you with a clearer understanding of our experimental conclusions. We are very grateful for your careful review of our research and your recognition of this study.

  1. What is the rationale for looking into apoptosis?

Response:

We think the points you raised are very specialized and research-worthy, and we are more than glad to answer them. As exhibited in our results, apoptosis is the most common form of programmed cell death in neuronal ischemia-reperfusion injury and can be triggered by intrinsic or extrinsic pathways. The initial morphological changes of apoptosis include cell shrinkage and cytoplasmic condensation, followed by nuclear membrane breakdown and apoptotic vesicle formation. Apoptosis can be viewed as cell death induced by activation of the Bax/Bak pore or cysteine asparaginase. Bax promotes apoptosis and Bcl2 inhibits apoptosis11. Possible mechanisms of neuronal apoptosis include 1. calpain-mediated apoptosis, 2. reactive oxygen species-mediated apoptosis, 3. DNA damage-mediated apoptosis, 4. inflammatory cells inducing apoptosis in the exogenous/death receptor pathway12. The present findings in paragraph 3.5 reveal that microglia Notch1 affects neuronal apoptosis induced by OGD/R, which is to further demonstrate that neurons can be further protected from hypoxia/reoxygenation injury by modulating microglia Notch1 expression, and that apoptosis is an important indicator of survival in response to neurons affected by OGD/R. Regarding the specific internal mechanism of microglia Notch1 affecting neuronal apoptosis needs to be further explored, and our study will further clarify its internal mechanism. Our current findings provide important preliminary clues and evidence support for future research in related fields.

  1. Bcl2 quantification in figure 5 does not match the representative image of the gel.

Response:

We thank the reviewers for meticulously suggesting revisions to this article, and we have scrutinized the representative images presented. To facilitate an accurate reflection of the findings of this experiment, we have updated the Western blot images of Bcl2, please refer to our latest uploaded manuscript (Figure 5E). We would like to express our sincere gratitude and appreciation for your review of this manuscript.

  1. Figure 6D is very hard to read.

Response:

Thanks for your questions. To investigate the molecular mechanism through which Notch1 in BV2 cells affects pyroptosis in HT22 neurons, we utilized the STRING database and Cytoscape software to construct the PPI network of Notch1, and the results indicated an interaction between Notch1 and JAK2/STAT3 (Figure 6D). Fig 6D, conducted by STRING database which (https://cn.string-db.org/) is an objective, extensive global network aimed at collecting, integrating and scoring the published protein–protein interaction (PPI) information, and at supplementing this data through scientific calculations and predictions, shows the interaction between Notch1, JAK2, STAT3. Similar article can be found in this reference13.

  1. In 6E, P-JAK2 data does not look significant.

Response:

Thanks for your suggestions. Since the P-JAK2 image in Figure6E was not representative enough, we updated the more representative Western blot image. After careful verification, it is consistent with the statistical results, please refer to our latest submitted manuscript. We would like to express our sincere appreciation for your careful review of the results data of this study.

  1. In the discussion, the M1/M2 discussion should be avoided.

Response:

The suggestions given by the reviewers were gladly accepted, and we have applied the revisions to the latest version of our manuscript. At the request of the reviewers, we avoided a large portion of the Discussion on microglia M1/M2, which we have abridged, as described in the second paragraph of the Discussion. We hope that the changes we have made to the manuscript will improve the presentation of this study.

  1. The discussion needs a more thorough description of what has been already shown about the inflammasome and stroke.

Response:

Thanks for your suggestions. We are more than glad to apply the proposals to the original article, and we have made exhaustive revisions, which can be found in the latest manuscript submitted (Line 532-536). We sincerely appreciate your review of this manuscript.

References:

  1. Campbell BCV, Khatri P. Stroke. Lancet. 2020;396:129-142
  2. Collaborators GBDLRoS, Feigin VL, Nguyen G, Cercy K, Johnson CO, Alam T, et al. Global, regional, and country-specific lifetime risks of stroke, 1990 and 2016. N Engl J Med. 2018;379:2429-2437
  3. Shi J, Gao W, Shao F. Pyroptosis: Gasdermin-mediated programmed necrotic cell death. Trends Biochem Sci. 2017;42:245-254
  4. Miao EA, Leaf IA, Treuting PM, Mao DP, Dors M, Sarkar A, et al. Caspase-1-induced pyroptosis is an innate immune effector mechanism against intracellular bacteria. Nat Immunol. 2010;11:1136-1142
  5. Bauernfeind F, Hornung V. Of inflammasomes and pathogens--sensing of microbes by the inflammasome. EMBO Mol Med. 2013;5:814-826
  6. Shi J, Zhao Y, Wang Y, Gao W, Ding J, Li P, et al. Inflammatory caspases are innate immune receptors for intracellular lps. Nature. 2014;514:187-192
  7. Wang G, Wang T, Zhang Y, Li F, Yu B, Kou J. Schizandrin protects against ogd/r-induced neuronal injury by suppressing autophagy: Involvement of the ampk/mtor pathway. Molecules. 2019;24
  8. Zhu H, Jian Z, Zhong Y, Ye Y, Zhang Y, Hu X, et al. Janus kinase inhibition ameliorates ischemic stroke injury and neuroinflammation through reducing nlrp3 inflammasome activation via jak2/stat3 pathway inhibition. Front Immunol. 2021;12:714943
  9. Wang J, He W, Zhang J. A richer and more diverse future for microglia phenotypes. Heliyon. 2023;9:e14713
  10. Zhong Y, Gu L, Ye Y, Zhu H, Pu B, Wang J, et al. Jak2/stat3 axis intermediates microglia/macrophage polarization during cerebral ischemia/reperfusion injury. Neuroscience. 2022;496:119-128
  11. D'Orsi B, Mateyka J, Prehn JHM. Control of mitochondrial physiology and cell death by the bcl-2 family proteins bax and bok. Neurochem Int. 2017;109:162-170
  12. Tuo QZ, Zhang ST, Lei P. Mechanisms of neuronal cell death in ischemic stroke and their therapeutic implications. Med Res Rev. 2022;42:259-305
  13. Jin W, Zhao J, Yang E, Wang Y, Wang Q, Wu Y, et al. Neuronal stat3/hif-1α/ptrf axis-mediated bioenergetic disturbance exacerbates cerebral ischemia-reperfusion injury via pla2g4a. Theranostics. 2022;12:3196-3216

Reviewer 3 Report

Comments and Suggestions for Authors

In this study, Chen et al. explored the influence of myeloid Notch1 on microglia polarization and neuronal pyroptosis in cerebral ischemia‒reperfusion injury (CIRI), aiming to uncover new therapeutic approaches for ischemic stroke. Utilizing myeloid-specific Notch1 knockout mice they assess the role of Notch1 in CIRI. The findings reveal that Notch1 activation occurs in cerebral myeloid cells post-CIRI, and its deficiency leads to reduced cerebral infarct volume, improved neurological function, and decreased peripheral monocyte/macrophage infiltration. Moreover, myeloid-specific Notch1 knockout notably mitigates pyroptosis and promotes microglia M2 polarization in the ischemic penumbra. Complementary in vitro experiments demonstrate that Notch1 knockdown in microglial BV2 cells curbs anoxia/reoxygenation-induced JAK2/STAT3 activation and pyroptosis in hippocampal neuron HT22 cells. These results significantly advance our understanding of the myeloid Notch1 signaling pathway's role in neuronal pyroptosis during CIRI, proposing that targeting myeloid-specific Notch1 could be a viable strategy for treating ischemic stroke.

The paper presents a compelling concept and the data effectively supports the authors' conclusions. However, I recommend several revisions before considering the manuscript ready for publication.

Major revision

1.     I would have suggested crossing the Notch1fl/fl mice with LysM-CreERT2 mice, which would lead to conditional KO, instead of constitutive KO. This approach would allow for the specific knockout of Notch1 in myeloid cells (via tamoxifen) just prior to TMCAO, ensuring that the observed protective effects are not influenced by developmental variances in the myeloid cell population of Notch1M-KOmice relative to wild-type mice. Recognizing that incorporating this new model for all experiments may be beyond the scope of this study, I instead recommend performing flow cytometry-based profiling of the myeloid cell population in healthy 8-week-old Notch1fl/fl, Notch1M-KO, and WT C57/B6 mice. This would validate the absence of differences among these groups. Additionally, the discussion should address the potential impacts of developmental Notch.

2.     I would like the authors to perform a Western blot for Notch1 and NLRP3 on OGD/R BV2 cells, integrating the results as new panels in Figure 1. This would serve as additional confirmation of the in vivo results shown in Figure 1D.

Minor revision

1.     Line 19 should be corrected to "we evaluated the"

2.     Line 61-62 – “These cytokines are released into the extracellular environment, thus causing toxicity to neurons.” What cytokines? Who is releasing them? The sentence should be rewritten addressing these questions, to improve clarity.

3.     Line 325 – “Immunofluorescence technology was conducted” should be replaced with “Immunofluorescence staining was performed”

4.     Line 351 – replacing “certificating” with “confirming”

Round 2

Reviewer 2 Report

Comments and Suggestions for Authors

N/A

Comments on the Quality of English Language

N/A

Reviewer 3 Report

Comments and Suggestions for Authors

I appreciate the authors addressing most of my concerns. However, my primary issue (Point 1), which I believe is a significant flaw in the study, remains unaddressed. It seems my point may not have been fully understood by the authors. Crossing the Notch1fl/fl mice with LysM-Cre mice instead of LysM-CreERT2 mice (https://www.jax.org/strain/031674), results in constitutive knockout of Notch1 in myeloid cells. This likely affects the development of these cells, introducing a significant confounding factor that undermines the study's findings. Merely showing PCR results confirming the absence of Notch1 expression in myeloid cells isn't sufficient - it only verifies the lack of Notch1 expression, without addressing the developmental implications of such constitutive knockout. The advantage of using LysM-CreERT2 is its ability to control the timing of Notch1 knockout using tamoxifen, ideally just before the experiments, to avoid the confounding effects of developmental changes. Understanding that creating this new mouse line and redoing the experiments might exceed the scope of this study, I proposed a feasible alternative experiment as a compromise. Hence, I maintain my initial stance and recommend publication contingent on the authors performing and incorporating flow-cytometry based immune profiling of the myeloid cell population in 8-week-old Notch1FL/FL, Notch1KO, and WT C57/B6 mice into the manuscript.
